# Semantic Modeling Approach Supporting Process Modeling and Analysis in Aircraft Development

**Junda Ma** [1], **Guoxin Wang** [1], **Jinzhi Lu** [2,*], **Shaofan Zhu** [3], **Jingjing Chen** [4] and **Yan Yan** [1]

1    School of Mechanical Engineering, Beijing Institute of Technology, Beijing 100081, China; mjd2015@sina.cn (J.M.); wangguoxin@bit.edu.cn (G.W.); yanyan331@bit.edu.cn (Y.Y.)
2    EPFL SCI-STI-DK, Station 9, CH 1015 Lausanne, Switzerland
3    COMAC BATRI, Beijing 102211, China; zhushaofan@comac.cc
4    School of Economics, Fudan University, Shanghai 200433, China; jingjchen@fudan.edu.cn
*    Correspondence: jinzhi.lu@epfl.ch

**Abstract:** With the increasing complexity of aircraft development programs, the development processes of aircraft and their subsystems are continuously becoming complicated, leading to the growing risks of development cost across the entire life cycle. In this study, we proposed a model-based systems engineering approach to support process modeling of aircraft development using a multi-architecture modeling language KARMA. Simultaneously, property verification and hybrid automata simulation were used to implement the static cost analysis of each work task and dynamic cost analysis of the entire development process. Finally, a development process model of aircraft avionics system was created using a case study, in which cost analysis is implemented by the KARMA language. From the result, we found that the KARMA language enables the integration of the process modeling with static and dynamic analyses of the development process in a multi-architecture modeling tool MetaGraph 2.0.

**Keywords:** MBSE; cost analysis; static analysis; dynamic analysis development process; KARMA language

## 1. Introduction

The continuous increase of system complexity of products, such as autonomous vehicles and intelligent transport systems, leads to growing challenges for managing the development processes across the entire life cycle of a system. When designing the development processes, multiple aspects must be considered: (1) Systems engineering life cycle and specific industrial standards are always used to define real development processes. The real aircraft development process should be used among all stakeholders of different domains. A unified and graphical description of the entire development process is a basic for communications. (2) Statistic analysis for each task during the development process is important to confirm that the cost of each task is lower than the criteria used to control the budget. (3) The dynamic performance of the development process is required to understand the cost and time consumption across the entire life cycle. This is useful when project managers expect to understand how the budget and duration are arranged for different phases.

The complex system development process involves stakeholders from different domains. Moreover, these stakeholders implement co-designing and collaborative designing across the entire life cycle from concept stage to retirement stage. Therefore, when designing the development process, a standardized specification is used as a basic for formalizing and tailoring the entire workflow in each domain to integrate different system interests of stakeholders. Except for systems engineering specifications, a model-based approach is also used to define business process, such as business process model and notation (BPMN) [1]. These

models enable stakeholders to understand their internal business procedures in a graphical notation and make organizations communicate these procedures in a standardized manner.

Before starting a complex system development process, cost and time consumption are two important aspects considered by stakeholders [2]. Low budgeting control for each work task can decrease the financial risk when the project is implemented. Moreover, the dynamic analysis of cost and time consumption for the development process provides the project managers better understanding of the project progress before the project implementation. Before implementing the project based on the designed development process, the static verification for cost and time in each work task is useful to understand if the task satisfies the requirements on the development process. The dynamic analysis provides a guideline for decision-makings on the development process to understand the dynamic performances of the entire project implementation.

The aim of this study is to propose a semantic modeling and simulation approach to support process modeling and analysis. The following contributions are introduced as follows:

- **Support the developmental process definition by semantic modeling:** A semantic modeling approach is proposed to provide a metamodel library for standardized development process development to improve the reuse of the existing work tasks when designing the development process. This approach provides a standardized syntax for constructing the graphical notation representing the development processes for all the stakeholders across the life cycle;
- **Support process analysis based on the semantic modeling:** Using the semantic modeling approach, static verification is used for stakeholders to validate if each task can satisfy the requirements related to cost for each work task. Moreover, for development process, dynamic analysis is implemented based on the semantic modeling approach to predict the dynamic performance of cost and time consumption across the entire life cycle.

In this study, we use the KARMA language to support the development process modeling and the static and dynamic analyses of time consumption and cost for complex system development [3]. Moreover, within a multi-architecture modeling tool *MetaGraph 2.0* (http://www.zkhoneycomb.com/; accessed on 8 February 2022 ), metamodels for the development process are developed based on ISO/IEEE 15288 and BPMN [4]. Furthermore, the satisfiability modulo theory (SMT) and the hybrid automata simulation (HAS) are used to support the static analysis of cost in each work task and the dynamic cost analysis of the entire development process.

The rest of the paper is organized as follows: Section 2 introduces the related works and our research methodology. Our proposed semantic modeling approach is illustrated in detail in Section 3. In Section 4, a case study is used to represent how the approach supports the process modeling and the analysis for an aircraft avionics development process. Section 5 presents the evaluation of the case study. Finally, this study is concluded in Section 6.

## 2. Related Work

The existing research has covered various aspects of semantic modeling for development process and process analysis based on model-based approaches.

### 2.1. Process Modeling Using MBSE

The development process plan and design are challenging because of the increasing complexity of the system development and the development process involves multiple stakeholders from different domains across the organizations [5,6]. Currently, model-based systems engineering (MBSE) was proposed by INCOSE as *"MBSE is the formalized application of modeling to support system requirements, design, analysis, and V&V activities beginning in the conceptual design phase and continuing throughout development and later life cycle phases"* [7]. Therefore, development process is one of the important perspectives of MBSE to manage

the complexity of the development processes. For example, business process model and notation [8], object-process methodology notations [9], and activity diagram in SysML and UML [10] are widely used to define and model the development process for supporting process design and analysis.

Several existing studies were proposed to bridge the MBSE with project management and development process management. A PRINCE2 framework was used to support project management using business process diagrams [11]. Using this framework, process models formalize the development process across the entire project. For the production service system, UML and BPMN were used to define the workflow of data analysis [12]. BPMN was used to support the decision-making for managing the automated production systems across organizations [1]. The BPMN models provide cues for decision-makings for supporting the organizational management. Except for graphical notations, semantic modeling is also widely used for development process formalism and analysis.

### 2.2. Semantic Modeling for Development Process

Currently, semantic modeling and ontology are widely used to support process modeling to address different domain specific challenges [13,14]. Compared with traditional MBSE approaches, the semantic modeling approach provides a formal semantic specification to support data interoperability across different stakeholders. Semantic modeling was used to develop process models and provide decision-makings in acute ischemic stroke treatment process using reasoning [15]. Using semantic techniques, process models decrease errors during the treatment of patients through the recommendations provided for the users. Semantic integration was implemented to support the manufacturing process integration across the organization [16]. Service discovery was implemented to support the manufacturing process automation by using semantic modeling techniques. Process modeling using semantic techniques is widely used for knowledge management [17].

One of the important features to semantic modeling is to integrate development process models with other domain specific information. For example, ontology was used to integrate business process models and other domain specific knowledge [18]. Semantic modeling was used for process identification using natural language processing. Ontology was designed to identify the development process from natural language [19]. Semantic techniques were used to integrate heterogeneous business process models using a unified ontology [20]. Therefore, a semantic approach, which integrates different language specifications for development process formalism, is important to the process integration of complex system development because of heterogeneous business process features and language specifications.

### 2.3. Process Analysis Based on Model-Based Approaches

Several existing analysis techniques are used for static and dynamic analyses of process features based on development process models. SysML and BPMN support development process formalism, which generate OWL models for reasoning, to provide decision-makings for project analysis [21]. NuSMV language is used to verify the process features based on LTL reasoning, which is generated from BPMN models [22]. Another semantic verification technique based on the Maude checker was used to support formal analysis of business process collaborations [23].

Except for the static verification, dynamic analysis of development process was implemented based on petri-net simulation that was used to support flexibility and performance analysis of development processes based on the BPMN models [24]. A colored petri-net approach was used to identify the business process with the time constraint [25]. A semantic language Promela was used to implement petri-net simulation that generated from BPMN models to verify the project performance of business process [26]. Although static and dynamic analyses are implemented based on graphical notation and semantic approaches separately, there is no integrated approach that combined the development process formalism and both the statistic and dynamic analyses.

### *2.4. Summary*

Based on the literature review, we find that semantic modeling supports the development process formalism and modeling. Moreover, it can integrate the domain specific knowledge and development process to help the project managers to identify the process information and system architecture. With analysis and reasoning, semantic development process models can provide decision-makings for process developers. Therefore, we identified several objectives in this study.

- A semantic modeling approach is proposed to support development process modeling, which can support different language specifications, such as activity diagram in SysML, UML, and BPMN;
- Syntax is extended in this semantic modeling approach to support static and dynamic verifications to evaluate process features and implement performance analysis.

### 3. Semantic Modeling Approach for Process Modeling and Analysis

We proposed a semantic modeling approach based on KARMA language to support process formalism and analysis and realize the research objectives. In this section, we introduce our entire approach, the relative KARMA formalism for development process, and the implementation of statistic and dynamic analyses.

### *3.1. Overview of the Semantic Approach*

The GOPPRR method proposed by Wang et al. is used to model reasoning through ontology, but the method lacks support for simulation analysis [27], so, based on their contribution, a semantic modeling language KARMA is developed based on the GOPPRRE approach, as shown in Figure 1. Using KARMA, metamodels are developed based on different process modeling specifications, such as BPMN or ISO/IEEE 15288 standards [4]. For the representation of the development process of complex systems, business process diagram (BPD) models are developed based on metamodels. The KARMA language formalizes the metamodels and models based on a unified specification. Moreover, through the KARMA language, property values are captured to define formal constraints of the BPD models based on SMT. Then, the SMT solver implements the static verification of development processes based on KARMA language. Furthermore, continuous system features for each task and discrete event system features are defined in each object instance and relationship instance, respectively, using KARMA language based on hybrid automata simulation (HAS). Finally, for dynamic analysis of development processes, simulation of process modeling is implemented by an HAS solver to provide results.

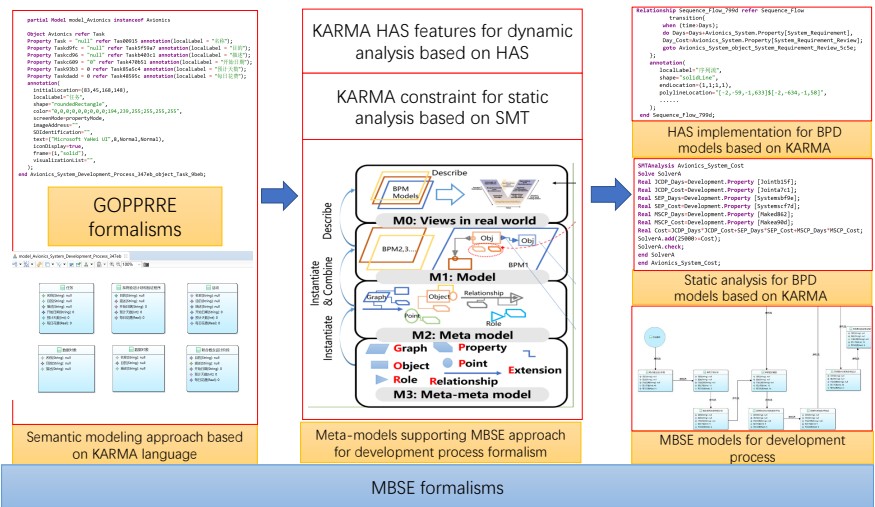

**Figure 1.** Overview.

### 3.2. KARMA Formalisms for Process Modeling

#### 3.2.1. GOPPRRE Formalism Supporting Metamodeling Development

As shown in Figure 2, the core of KARMA formalism is a GOPPRRE formalism for developing MBSE models, which is designed based on a M0-M3 modeling framework that supports metamodel and model development. Four layers are used to represent the development process using the semantic KARMA language to formalize the development processes using KARMA formalism.

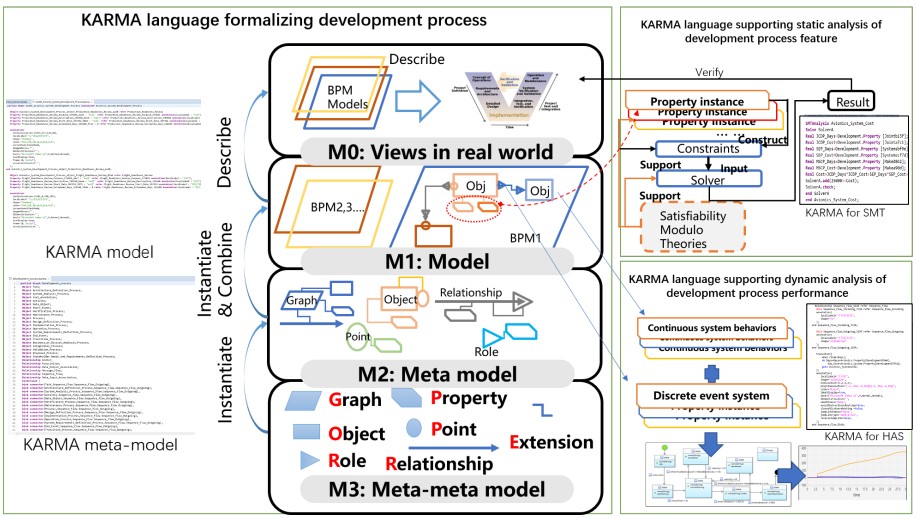

**Figure 2.** KARMA formalism and analysis for the development process modeling.

- M3 refers to meta-metamodels, including the basic elements for developing metamodels based on the core GOPPRRE concepts: (1) Graph; (2) Object; (3) Point; (4) Property; (5) Role; (6) Relationship ; and (7) Extension;
- M2 refers to metamodels used for constructing models. In this study, metamodels are used for constructing the BPD models, whose syntax and semantics are defined based on the graphical notation specification BPMN and industrial standards such as ISO/IEEE 15288, respectively;
- M1 refers to BPD models which describe the development process using graphical notations;
- M0 refers to the real development process of the aircraft avionics system.

Based on the *GOPPRRE* formalism, the syntax of KARMA languages, which is one of the most powerful approaches to describe domain specific metamodels of products [28], is used to describe metamodels and models. The detailed key concepts are introduced as follows:

- *Graph* refers to a collection of *Object*, and *Relationship* is represented as one window referring to a BPD model with graphical notations. To represent the hierarchy between different development processes, one *Object* should be decomposed into another BPD diagram graph model;
- *Object* refers to one entity in *Graphs* (one work task in a BPD model);
- *Point* refers to a port in each *Object*;
- *Relationship* refers to one connection between the two *Points* of *Objects* or *Objects* in a *Graph*;
- *Role* refers to each end of *Relationship* used to define the connection rules for the relevant *Relationship*. For example, one *Relationship* has two *Roles*. Each is defined to connect one *Point* in *Objects* or one *Object* with *Relationship*. Then, the connector between *Relationship* and the *Points* or *Objects* is created as a constraint to implement connections among different *Objects*;
- *Property* refers to one attribute in the other five non-property metamodels.

To formalize the BPD models for development process, a definition is proposed as follows:

The $::= ()$ refers to a collection. $Graph^a_{bpd}$ is defined as a BPD model $a$ based on the metamodel of Graph $bpd$. In $Graph^a_{bpd}$, $Object^b_{obj}$ refers to the object instance $b$ based on a metamodel of Object $Obj$, such as work task $b$. $Relationship^c_{Rel}$ refers to a relationship instance $c$ based on the metamodel of Relationship $Rel$, such as sequence from one work task to another. $Role^d_{Rol}(x)$ refers to a role instance $d$ based on a metamodel of Role $Rol$ in a relationship instance $x$, which is developed based on metamodel Relationship $Rel$; $Property^f_{Proi}(z)$ refers to the property instance $f$ based on the metamodel of Property $Proi$ in the model instance $z$, whose metamodel is $nonprop$ ($nonprop \subseteq \{Gi, Obji, Rei, Roi\}$). The formalism of a whole BPD model is shown in Equation (1).

$$Graph^a_{bpd} ::= (\sum Object^b_{Obj}, \sum Relationship^c_{Rel},$$
$$\sum Role^d_{Rol}(x, Rel), \sum Property^f_{Proi}(z, nonprop)) \tag{1}$$

One relationship instance $c$ is defined as $connector\_Instance(a)$ pointing to ($\Rightarrow$) $connector\_Instance(b)$. The $connector\_Instance(a)$ refers to a collection of an Object instance (work task) and a Role instance as the start point of $c$. The $connector\_Instance(b)$ refers to a collection of an object instance and a role instance, which is the end of $c$. Thus, the formalism of a connection in an BPD model is shown in Equation (2).

$$Relationship^c_{Rel} ::= (connector\_Instance(a) \Rightarrow connector\_Instance(b)). \tag{2}$$

### 3.2.2. Metamodels Supporting Process Modeling

The ISO/IEEE 15288 standard provides a common process framework covering the life cycle of complex systems. It is necessary to standardize the work task nodes as metamodel objects to realize the construction of aircraft process models and satisfy the development process of the aircraft avionics system. Through iterative modeling and analysis, a solution that balances key performance indicators, such as cost and time, is ultimately created. BPMN provides a standard notation that is easy to understand for all business stakeholders, including basic elements such as flow objects, connecting objects, artifacts, and swim lanes, and support for technical and business personnel to engage in business process management. Based on the GOPPRRE formalism, a BPD graph is created with references to 15,288 and the BPMN notation specification to support process modeling of the aircraft avionics system.

As shown in the Table 1, it contains metamodels of the life cycle technology management process and the technology process of the system life cycle. The main purpose of the life cycle technology management process is to build a technical baseline management process for implementing the decision gates at each stage, including nine review processes, such as preliminary design review, system function review, and system verification review. The technical process includes the development process stages of aircraft avionics system: concept stage, development stage, and production stage. In the concept stage, there are eight work tasks, including system engineering program, joint conceptual design, system requirements analysis, and architecture design, and subsystem requirements definition, to support the generation of system architecture solutions. In the production stage, there are 11 work tasks, including subsystem development, system integration, system verification, and system security assessment, to produce, integrate, test, and deliver the system. Therefore, considering each work task as a BPD object, a total of 33 BPD objects are created to support the instantiation of the corresponding work tasks. Simultaneously, five objects, including tasks, activities, processes, data objects, and text annotations, are created as general modeling elements to deal with work tasks involved in the avionics system work task modeling. Two objects, the start event and the end event, are created to show the start and end nodes of process modeling. Therefore, 40 object elements are constructed totally to fully describe the aircraft avionics system life cycle development process. Similarly,

according to the BPMN specification, six relationships are created, including connection annotation, association, data input association, data output association, information flow, and sequence flow, to express the object connection relation of BPD. In addition, 12 corresponding roles are also created. According to the constraints of objects and relationships, 427 connectors are constructed to indicate each binding. Due to the reusability of the property meta-model, only six basic property metamodels need to be constructed, namely name, purpose, description, start date, the estimated number of days, and daily cost, which can satisfy the addition of each object metamodel. The detailed object metamodels are shown in Table 2.

### 3.3. Process Analysis Based on KARMA Formalism

Based on the semantic modeling approach, static analysis was used to verify that the parameter values of the work tasks in the process model can satisfy the specified value constraints, and dynamic analysis was executed to predict the dynamic performance of cost and time consumption across the entire life cycle.

#### 3.3.1. Static Analysis for Process Models

Based on the GOPPRRE modeling method, the model elements were formalized by the multi-architecture language KARMA. With the KARMA features, satisfiability modulo theories (SMT) were used to support the static analysis of process models, which refers to a checking process where the satisfiability of logical formulas over one or more theories in the models combines the problem of Boolean satisfiability with some of the most fundamental fields in computer science. In addition, it draws on the most prolific problems in the past of symbolic logic: the decision problem, completeness and incompleteness of logical theories, and complexity theory [29]. The GOPPRRE method extended with SMT is used to realize the verification of constraint attributes in the process model and to test whether the logical constraints based on a situation of one or more mathematical theories are satisfied. Based on the corresponding solver of SMT, the property instances are formalized as constraints for first-order logical expression and evaluation.

Based on the basic MBSE formalism in KARMA, static cost analysis is implemented for each work task based on SMT solver in MetaGraph 2.0 [30]. As shown in Figure 3 , Meta-Graph supports metamodel development and modeling based on the GOPPRRE modeling method, where various work tasks based on the design requirements of the development process are constructed using the BPD metamodel for verification implementation. Then, KARMA language is used to formally describe the constraints of property instances to verify based on the defined constraints according to design requirements related to development process. Finally, the KARMA language is compiled by constructing and traversing the abstract syntax tree, which is used to verify if the constraints are satisfied by executing the SMT solver. The static analysis results, whether the constraints are satisfied, are transmitted to the result perspective in MetaGraph and demonstrated to the stakeholders.

The data structure tree of the extended KARMA language for static analysis is shown in Figure 4. The *propertyInstance* node is under the object class in the model layer, and is an instantiation of the *property* metamodel. In addition, the extended classes *SMTAnalysis* and *PropertyState* are shown in Table 3. Using such syntax, constraints for static analysis of the process model are defined and evaluated.

**Table 1.** Object metamodels for process modeling.

| Phase | Subphase | Process Mate Objects | Quantity |
|---|---|---|---|
| Technical management process | Technical baseline management | System requirements review, preliminary design review, system function review | 9 |
| | | Critical design reviews, production readiness reviews | |
| | | System validation review, validation readiness review, system qualification review | |
| Technical process | Concept | System requirements capture, requirements of system integration verification and experimental environment, subsystem requirements definition | 8 |
| | | Joint conceptual design phase, system requirements analysis and architecture design | |
| | | Requirements management, systems engineering management, systems engineering plan | |
| | Development | Function analysis and allocation, system design analysis, subsystem design | 5 |
| | | System validation plan and validation procedure, system core integration and validation plan | |
| | Production | Subsystem validation, Subsystem manufacturing, Subsystem certification | 11 |
| | | System integration, system verification, system safety assessment, system certification plan formulation, system integration verification test environment development | |
| | | System FHA and preliminary PSSA, system security plan and preliminary FHA, system assessments of reliability, maintainability and availability | |
| General | | Tasks, activities, process, data object, text annotation | 5 |
| Others | | Start event, end event | 2 |
| Total | | | 40 |

**Table 2.** Metamodel elements included in a BPD.

| Graph | Object | Relationship | Point | Role | Property | Extension (Connector) |
|---|---|---|---|---|---|---|
| 1 | 40 | 6 | 0 | 12 | 6 | 427 |

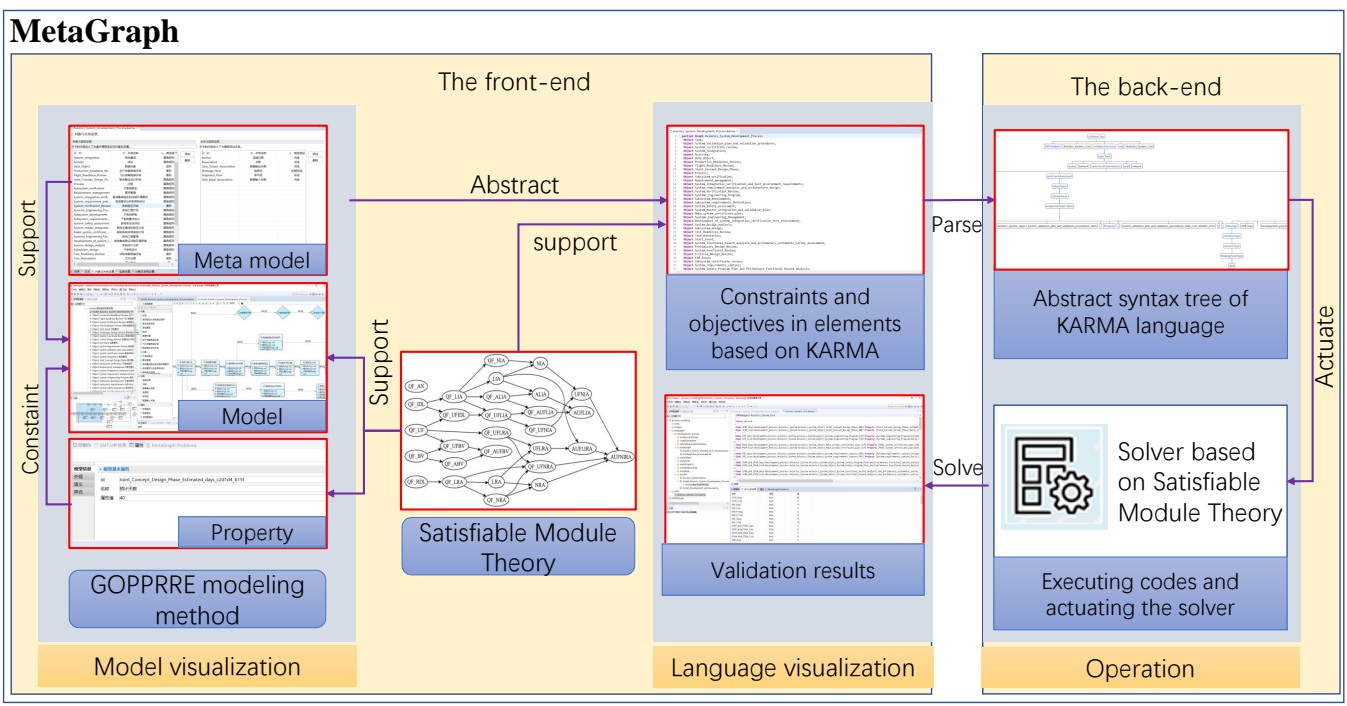

**Figure 3.** Implementation of static analysis for the process model.

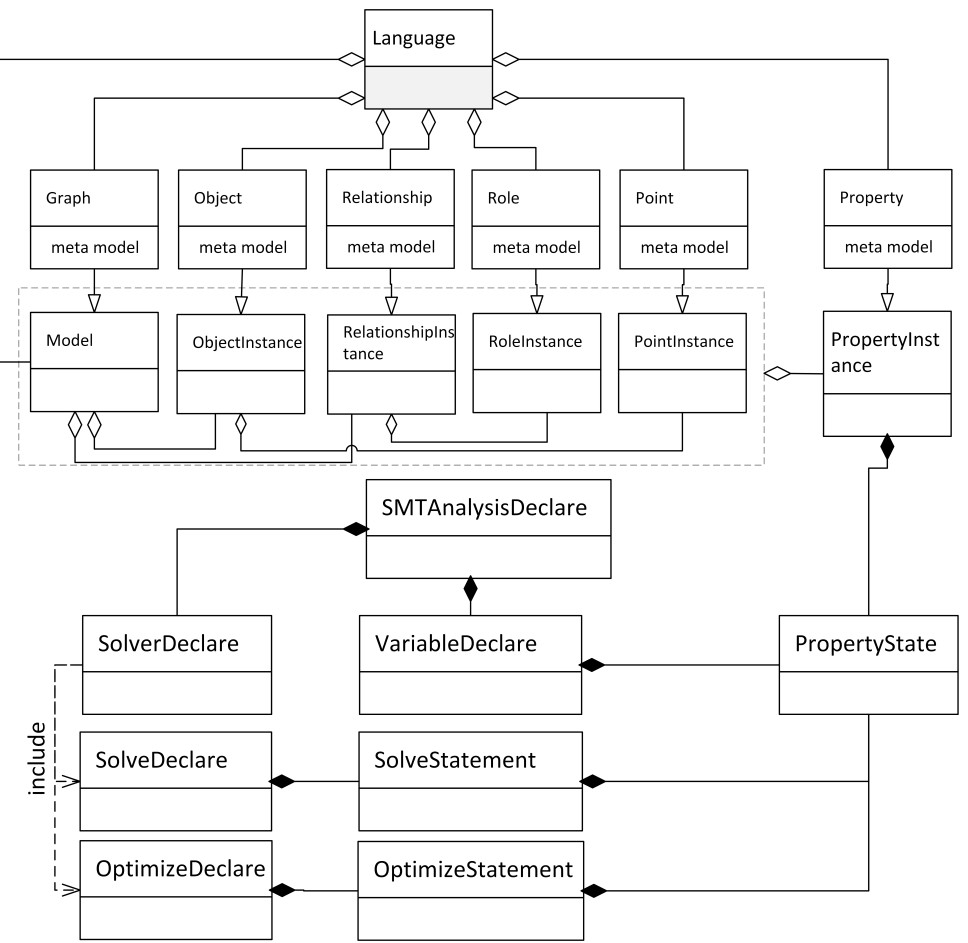

**Figure 4.** Data structure tree of the extended KARMA language for static analysis.

**Table 3.** Syntax of KARMA Language for static analysis.

| Class | Description | Syntax |
| --- | --- | --- |
| SMTAnalysis | Declare the module of analysis for property verification. | SMTAnalysis <Name> ... end <Name> |
| PropertyState | Capture the value of property instance. | languageID. modelID. ObjectID. Property[PropertyID] |
| VariableDeclare | Declare the variables. | Int a; Int b = valueOf(); Boolean x; Boolean y = true; Matrix m = IntegerMatrex; Matrix n = [1,1;2,2]; ... |
| SolveDeclare | Declare that the property verification type is to evaluate the satisfiablity of constraints. | Solve <Name> ... end <Name> |
| OptimizeDeclare | Declare that the property verification type is to optimize the variable value of property instances. | Optimize <Name> ... end <Name> |
| solveStatement | Indicate the statements of how to evaluate the constraints. | <Name>.add(a&&(~b,...)) <Name>.push; //construct a temporary stack; <Name>.check; |
| optimizeStatement | Indicate the statements of how to optimize the constraints. | <Name>.push; <Name>.pop; <Name>.Max(a+b); <Name>.solution; |

### 3.3.2. Dynamic Analysis for Process Model

Hybrid automaton refers to a finite state machine with continuously evolving variables and discrete jump transitions to describe the dynamic behavior of a system. Ding et al. attempted to describe the hybrid automation using KARMA language in the method level to support dynamic analysis [31]. On the basis of their method, we use it in the practice of process modeling. In our research, the continuous system state and discrete event system features in the process model for verification are defined. The continuous system state describes the variables of the parameter state in the system evolution with time and events, for example, vehicle speed and mileage are physical quantities that change continuously with time [32]. The discrete event system refers to a dynamic system that is driven by events and changes the state of the system by leaps and bounds [33]. The behavior of discrete systems is generally described by formal models such as logic and algebra.

The hybrid automaton is a generalized finite-state automaton that is furnished with continuous variables. The syntax of hybrid automata is expressed as [34]

$$H = < Loc, Edg, \Sigma, X, Init, Inv, Flow, Jump >$$

- $Loc$ is a finite set $\{l_1, l_2, ... l_n\}$ of system discrete states;
- $\Sigma$ is a finite set of event names;
- $Edge \subseteq Loc \times \Sigma \times Loc$ is a finite set of labeled edges which represent discrete changes;
- $X$ is a finite set $\{x_1, x_2, .. x_m\}$ of real-valued variables;

- *Init*, *Inv*, *Flow* are functions that assign three predicates to each location. *Init* and *Inv* define sets of initial states and invariants that set value constraints for each state, respectively. *Flow* is a set of predicates that defines the continuous evolution of the hybrid system in one state;
- *Jump* is a function assigning to each labeled edge a predicate that provides discrete events when a hybrid automaton moves from one state to another.

In the KAMRA language, the mapping expression of the hybrid automaton *H* is realized, in which *Loc*, *Init*, *X* and *Inv* are mapped to the object metamodels, and *Edg*, *Flow*, *Jump*, and Σ are mapped to the relationship metamodels.

The process model with various state valuables and discrete event features are constructed according to the design requirements in MetaGraph based on the metamodel of BPD to perform hybrid automata simulation, as shown in Figure 5. Based on the hybrid automata theory, the functions and variable declarations supporting hybrid automata simulation are also defined with property values representing the state within the BPM models simultaneously. Then, the BPM models with hybrid automata syntax are interpreted as the formal abstract syntax tree from KARMA language. Subsequently, the BPM models based on KARMA language is compiled to generate CIF specifications to execute the hybrid automata simulation by a CIF solver [35]. Finally, after hybrid automata simulation is executed, simulation results are generated to verify the dynamic performances of the development process.

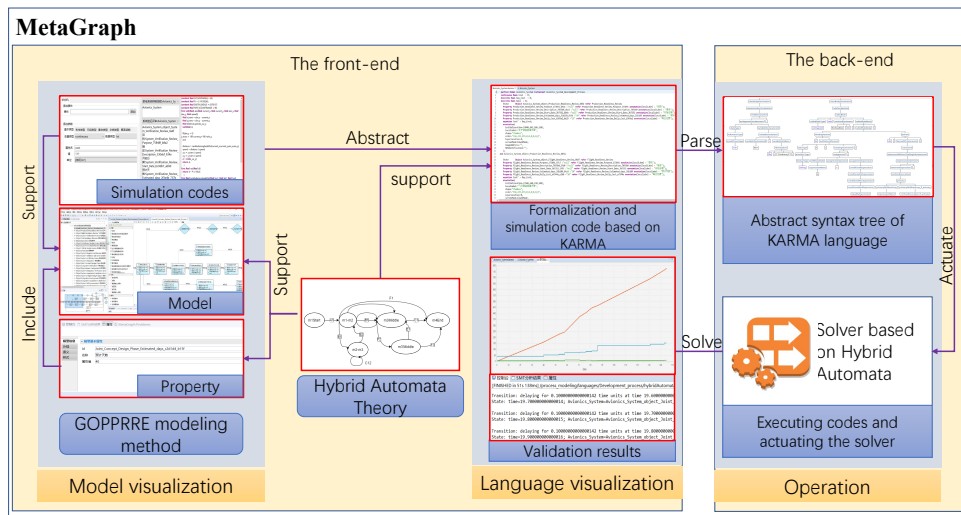

**Figure 5.** Implementation of the hybrid automata simulation for the process model.

The data structure tree of the extended KARMA language for hybrid automata simulation is shown in the Figure 6. The *propertyInstance* node is under the *Object* class in the model, which is an *instantiation* of the property metamodel. The detailed syntax of the KARMA extended classes *StateObjectInstance* and *PropertyState* are shown in Table 4.

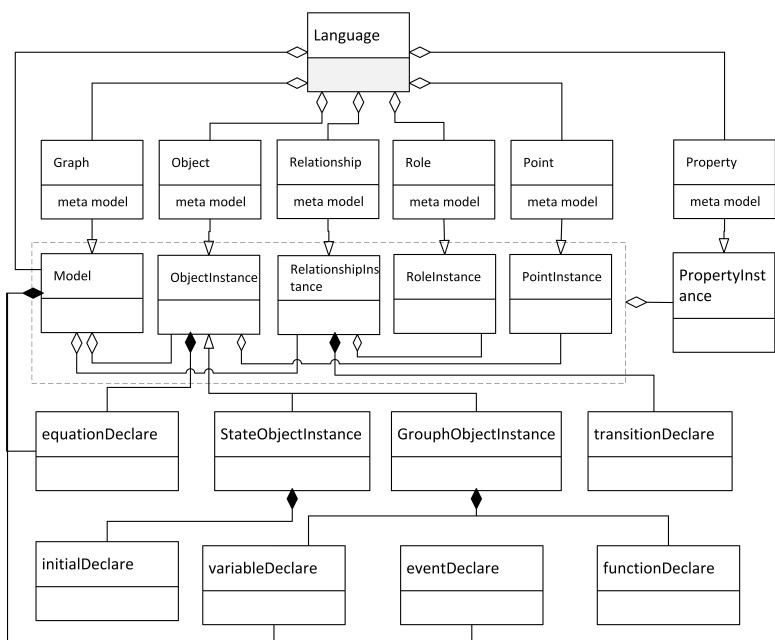

**Figure 6.** Data structure tree of the extended KARMA language for hybrid automata simulation.

**Table 4.** Syntax of KARMA language for hybrid automata simulation.

| Class | Description | Syntax |
|---|---|---|
| StateObject-Instance | Define object instances of the State type. | State Object<ID>refer<metaID><br>....<br>end <ID> |
| GroupObject-Instance | Define object instances of the Group type. | Grouph Object<ID>refer<metaID><br>....<br>end <ID> |
| initial-Declare | Identify the initial State in object instances. | initial; |
| variable-Declare | Declare variables including discrete, continuous, constant, and algebraic types. | discrete Boolean a = false;<br>continuous Real b = 1.12;<br>algebraic Int c = fun(x);<br>constant Int d = 10; |
| eventDeclare | Declare events. | event<enentName>; |
| transition-Declare | Declare the transitions of relationship instances containing guards, effects and new states. | transition {<br>[event ;]? %guard<br>[when();]? %guard<br>[do ;]? %effect<br>goto ; %new state }; |
| equation-Declare | Declare equation to describe behaviors in Object or Graph instances. | equation t = 2; |
| function-Declare | Declare functions that can solve complex logical and mathematical calculations. | func<returnType><funcName><br>(Int 2)<br>{<funcBody>} |

## 4. Case Study

### 4.1. Problem Statement

The aircraft avionics system is a key system to uniformly control various avionics components on the aircraft, transmitting relevant information between the avionics compo-

nents through the on-board data bus. As the development process of the avionics system becomes more and more complex, the development, installation, and commissioning of different avionics affect the life cycle cost of the whole aircraft. Therefore, its cost and time consumption will lead to increased risk of R&D costs throughout the life cycle. The process control for the development process of the aircraft avionics system in the early R&D stage has become the primary task before project implementation. The process model of the avionics system development was built based on the KARMA language, in which both static and dynamic analyses are based on SMT and hybrid automata simulation Here, the static analysis analyzes whether the overall cost of the avionics system work tasks is within the constraints of the target cost, while the dynamic analysis analyzes the dynamic changes in the stacking of various work task nodes in terms of time and cost.

### 4.2. Evaluation Criteria

In the case study, the BPD Graph was used to develop the development process of the aircraft avionics system development. A process model was developed using the BPD metamodels, as shown in Figure 7, which includes key work tasks for the entire process, such as start event, systems engineering plan task, system design analysis task, system security assessment task, critical design review task, and end event task.

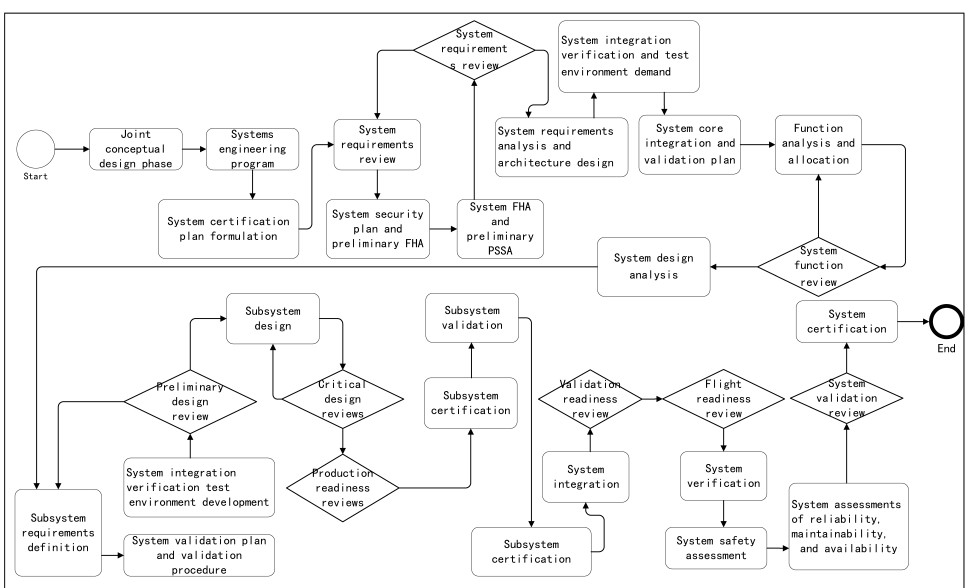

**Figure 7.** Aircraft avionics system process model.

Based on the process model and related KARMA language, the static analysis of the cost for each work task in the process model is defined as follows:

$$\sum (t_{task} \times Cost_{task}) \in Constraint(Static\_Cost\_E) \tag{3}$$

where $t_{task}$ refers to the days for each work task, $Cost_{task}$ refers to the cost for each day in one work task (e.g., the days of **system engineering program** is 13 and its cost of each day is CNY 50,000 , as shown in the Figure 8, $SEP\_Days = 13$, $SEP\_Cost = 5$ (CNY 10,000)). $Constraint()$ refers to the constraint function, $Static\_Cost\_E$ refers to variables to construct constraint functions (in this case, $Static\_Cost\_E$ is the total target cost CNY 250,000,000), and $\in$ refers to the $t_{task} * Cost_{task}$, which is required to satisfy the constraint. To achieve this, we used the KARMA code $Solver A.add(25000 >= Cost)$.

```
Real SEP_Days=Systems_Engineering_Program.Property [Systems_Engineering_Program_Estimated_days];
Real SEP_Cost=Systems_Engineering_Program.Property [Systems_Engineering_Program_Daily_Cost];
...
Real Cost=SEP_Days*SEP_Cost+JCDP_Days*JCDP_Cost+...;
SolverA.add(25000>=Cost);
SolverA.check;
end SolverA
```

**Figure 8.** KARMA language for analyzing the static cost

The dynamic analysis of the cost in the process model is shown as follow:

$$Dynamic\_Cost_{process} = \sum_{i}^{n}(T_{im} \times Cost_i) \qquad (4)$$

where $Dynamic\_Cost_{process}$ refers to the dynamic cost of the entire process model since time is increasing, $i$ refers to the number of the work tasks, $T_{im}$ refers to the number of days to perform the i-th work task, $Cost_i$ refers to the cost of each day in the i-th work task. The cost and state transition of **system engineering program** are shown in the Figure 9, $Day\_Cost$ represents the dynamic cost of **system engineering program** node.

```
equation Cost' = Day_Cost;
 transition{
  when (time>Days);
  do Days=Days+Systems_Engineering_Program.Property[c2d1d4_bf9e],
    Day_Cost=Systems_Engineering_Program.Property[c0ea26_cf7d];
  goto Systems_Engineering_Program;
};
```

**Figure 9.** KARMA language for analyzing the dynamic cost.

Based on the process model, both the static and dynamic analyses of the process model were implemented. Figure 10 presents the results, which help systems engineers and project managers in managing their projects and the development process before the implementation of these projects.

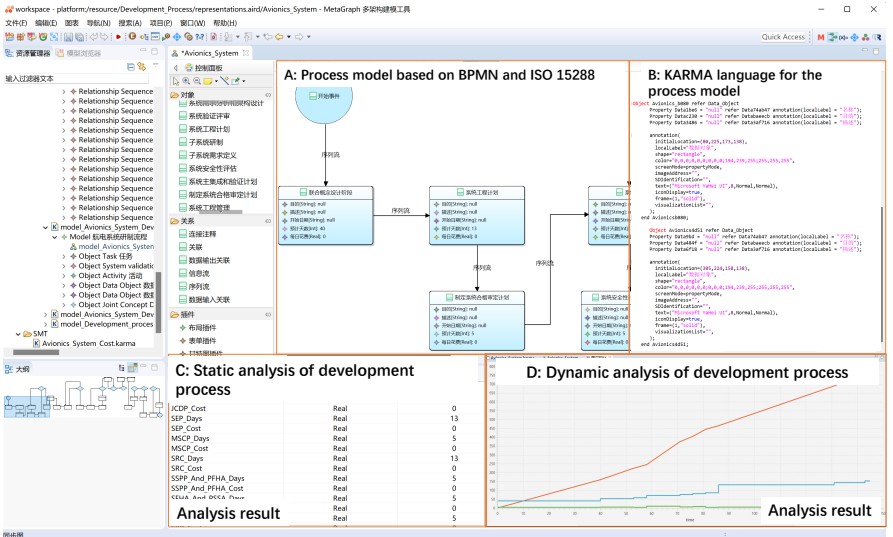

**Figure 10.** KARMA language supporting the process model development and cost analysis.

The cost for each work task in the process model is demonstrated in Table 5. An example of static analysis of the task cost was presented. Based on the results, we find that the cost in the mission analysis task satisfies the constraint in Figure 10C. On the other hand, from the result of dynamic analysis, we find the dynamics for the whole process model in Figure 10D.

**Table 5.** Properties for each work task in the KARMA model.

| Dynamic Analysis | | | | |
|---|---|---|---|---|
| **Work Task** | **Work Task Cost for Each Day (CNY 10,000)** | **Planning Time for Each Work Task** | **Process Cost (CNY 10,000)** | **Cost for the Entire Process Model (CNY 10,000)** |
| Joint concept design | 4 | 40 | 160 | |
| Systems engineering program | 5 | 13 | 65 | 245 |
| System certification plan formulation | 4 | 5 | 20 | |
| System requirements review | 10 | 13 | 130 | |
| System security plan and preliminary FHA | 6 | 5 | 30 | |
| System FHA and preliminary PSSA | 8 | 5 | 40 | 465 |
| System requirements review | 4 | 5 | 20 | |
| System requirements analysis and architecture design | 5 | 45 | 225 | |
| System integration verification and test environment demand | 6 | 12 | 72 | 842 |
| System core integration and validation plan | 8 | 10 | 80 | |
| Function analysis and allocation | 10 | 50 | 500 | |
| System function review | 15 | 10 | 150 | |
| System design analysis | 23 | 60 | 1380 | |
| Subsystem requirements definition | 18 | 10 | 180 | |
| System validation plan and validation procedure | 16 | 15 | 240 | 12,627 |
| System integration verification test environment development | 15 | 30 | 450 | |
| Preliminary design review | 20 | 7 | 140 | |
| Subsystem design | 35 | 243 | 8505 | |
| Critical design reviews | 12 | 20 | 240 | |
| Production readiness reviews | 15 | 10 | 150 | |
| Subsystem certification | 55 | 145 | 7975 | |
| Subsystem validation | 30 | 30 | 900 | 21,952 |
| Subsystem certification | 20 | 15 | 300 | |
| System integration | 15 | 80 | 1200 | 23,152 |
| Validation readiness review | 8 | 10 | 80 | |
| Flight readiness review | 10 | 10 | 100 | |
| System verification | 15 | 50 | 750 | |
| System safety assessment | 10 | 7 | 70 | 24,456 |
| System assessments of reliability, maintainability, and availability | 8 | 8 | 64 | |
| System validation review | 12 | 20 | 240 | |
| System certification | 4 | 20 | 80 | 24,536 |
| Static analysis | | | | |
| Verification item | Cost for all work tasks (CNY 10,000) | Constraint (CNY 10,000) | Result | |
| Cost of business or mission analysis work tasks | 24,536 | <25,000 | The property satisfies the constraint | |

## 5. Discussion

### 5.1. Quantitative Analysis

In the case study, the semantic approach was used to define the development process of the aircraft avionics system. The formalism of the graphic models constraints and calculations of static analysis were defined to verify if the cost in each work task satisfies the constraints based on KARMA language. Based on the KARMA model, SMT execution was performed to obtain an acceptable task cost based on the design requirement. In addition, the state, group, and event and transition for dynamic analysis are added to the KARMA model to simulate the dynamic changes in costs.

In Table 6, we presentsthe quantitative analysis for the KARMA model, including the numbers of KARMA elements in models and KARMA language lines. The BPD model, named *The overall development process model of aircraft avionics system*, was used to define the development process, which includes 2 event nodes, 8 review nodes, and 23 development nodes. There are 1875 lines of KARMA language describing the process model. The 23 development process objectives have the properties of days and costs. Finally, for performing static analysis, 104 lines of KARMA language were used to describe the constraints and objectives of the development process. In addition, 138 lines of KARMA

language were used to describe the continuous variables and algebraic variables and events to support dynamic analysis.

**Table 6.** Quantitative analysis of the case.

| Quantitative Analysis of Models | | |
|---|---|---|
| **GOPPRRE** | **Instances** | **Quantity** |
| Graph | The overall development process model of the aircraft avionics system. | 1 |
| Object | Start event, systems engineering plan, etc. | 33 |
| Relationship | Sequence flow_1, sequence flow_2, etc. | 36 |
| Point | none | 0 |
| Role | FlowInput_1,FlowOutput_1, etc. | 72 |
| Property | Estimated days, daily cost, etc. | 115 |
| Extension (connector) | connector (Avionics_System_object_System_certificate_review_d1a3, Sequence_Flow_2be8.Sequence_Flow_Incoming_f50f),etc. | 72 |
| Quantitative analysis of the KARMA script | | |
| Formalized models in KARMA | | 1875 lines |
| Porperty verification in KARMA | | 104 lines |
| Hybrid automata simulation in KARMA | | 138 lines |

## 5.2. Qualitative Analysis

From a qualitative perspective, compared with the traditional avionics system development process planning using documents, the given approach can describe the development process of the avionics system and provide decision-making options for project managers by implementing the static and dynamic analyses. The KARMA language is a textual and semantic modeling language with meta-metamodeling capability developed based on the GOPPRRE meta-metamodels. It supports the construction of metamodels for development process from different domains and process modeling specifications with strong model description capability and expansibility. In this study, we built a BPD model that supports the process description of the aircraft avionics system, satisfying the product life cycle design process of the 15,288 specifications, such as tasks of system design and system requirements definition. However, it describes symbols in the BPMN standard, such as nodes of start event and end event, providing a consistent model building and communication environment for avionics developers.

The constraints defined based on SMT were formalized involving complex mathematical theories, providing results in Boolean as returning. These results can help project managers to understand if each work task can satisfy the constraints defined based on the given requirements. Using the KARMA syntax, the process model was built first. Then, costs and days related to cost analysis were defined in a unified formalism. The verification of the satisfaction related to the cost of the avionics system development process was implemented automatically using the SMT solver by defining the cost constraints using the KARMA code. This approach helps stakeholders gain a unified understanding and representation of the process using the graphical notations and enables stakeholders to monitor the satisfaction of plans and objectives when defining processes.

A hybrid automaton is a finite state machine with discrete jump transitions and continuous variables. The dynamics of the entire development process can be calculated by using the finite state machine that represents a finite number of states and a mathematical model of behavior, such as transitions and actions between those states [36]. In this study, KARMA language was realized to express the group, equation, transition, and other state attributes based on the hybrid automata theory. We defined the cost equations and variable

values in the object nodes. In addition, transitions were defined in the relationships, such as day stacking and cost accumulation. Finally, cif solver was used for the simulation. From the simulation results, stakeholders can adjust the task process through the dynamic quantitative data of cost and predict the dynamic performance of cost and time consumption throughout the life cycle.

*5.3. Summary*

The KARMA language based on the GOPPRRE metamodeling method was proposed to describe the development process of aircraft avionics systems. In the early stage of designing the development process, static analysis was used to analyze whether the R&D cost of the avionics system meets the requirements to help the stakeholders of the system development quickly realize the requirement verification. On the other hand, through dynamic analysis, the cost curve of the avionics system R&D process was analyzed to help stakeholders optimize the defined plan. The specific features of the proposed modeling approach are summarized as follows:

- The KARMA language is a semantic modeling language with GOPPRRE concepts, which is designed based on a M3-M0 modeling framework supporting metamodels and model development. To formalize the development of the R&D process, we develop a metamodel Graph called BPD in consideration of the BPMN modeling specifications and the ISO/IEEE 15288 life cycle R&D process standards for the process development of aircraft avionics system based on the KARMA language;
- The static analysis method based on the KARMA language was implemented based on a unified formal expression of the satisfaction modulo theory. The method supports the construction of first-order logic constraints according to requirements and realizes the satisfaction verification of target cost using cost analysis;
- The dynamic analysis based on the KARMA language was implemented based on a unified formal expression of hybrid automata theory, which represents continuous dynamic systems by adding a set of differential equations to the state of the automaton [37]. Dynamic analysis supports the representation of the dynamic features of the development process through simulation.

## 6. Conclusions

This study proposed an MBSE approach for process modeling and cost analysis using KARMA language. First, KARMA language was introduced to support process modeling using a GOPPRRE approach. Second, based on KARMA language, static and dynamic analyses of development processes of aircrafts were implemented by using property verification and hybrid automata simulation. Finally, using the case study, we found that the KARMA models describe a process model of the aircraft development processes and support cost analysis not only from the static perspective but also from the dynamic perspective. Using this approach, the development processes can be analyzed during project planning, enabling the decrease of the risks of project failures because of cost overrun. The contributions are summarized as follows:

- A semantic KARMA language supports development process modeling based on the ISO/IEEE 15,288, BPMN, and other specific development process features, which improves the consistency of the process model in the aircraft system development;
- Syntax is extended in KARMA language enables to support static verification and dynamic simulation using SMT and hybrid automata simulation to evaluate process features and implement performance analysis of development process, thereby improving the accuracy of the models of the system R&D early stages;
- A case study regarding the entire life cycle modeling and verification was implemented based on the proposed approach. From the case study, the work task was formalized using the semantic approach and its static features and dynamic performances were evaluated by the solvers.

For future work, it would be very interesting to develop more metamodels of other process modeling specifications based on the KARMA language to define more complex system development processes.

**Author Contributions:** methodology, J.M. and J.L.; modeling, J.M.; static analysis and dynamic analysis, J.M.; writing—original draft preparation, J.M. and J.L.; writing—original draft preparation, J.M., J.L., G.W., J.C., S.Z. and Y.Y.; supervision, J.L. and Y.Y; funding acquisition, G.W.; All authors have read and agreed to the published version of the manuscript.

**Funding:** This study is supported by the CN National Key Research and Development Plan (Grant No.2020YFB1708100)and the CN pre-study common technology(50923010101).

**Data Availability Statement:** We have produced detailed video material for the case of aircraft avionics system, and the video is available at https://youtu.be/mijjX0Atag0 (accessed on 16 February 2022). We have uploaded all the code for modeling and analysis at https://gitee.com/zkhoneycomb/open-share.git (accessed on 16 February 2022)/Papers / Semantic Modeling Approach Supporting Process Modeling and Analysis / process_modeling.

**Institutional Review Board Statement:** Not applicable.

**Informed Consent Statement:** Not applicable.

**Data Availability Statement:** Not applicable.

**Conflicts of Interest:** The authors declare no conflict of interest.

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
