# Peer review of "Semantic Modeling Approach Supporting Process Modeling and Analysis in Aircraft Development"

_applsci, doi:10.3390/app12063067_

Round 1

Reviewer 1 Report

The organization of the paper is very good. The literature review provides adequate information regarding the topic. Authors propose an interesting MBSE approach for process modeling and cost analysis using KARMA language.

The case study proofs that the KARMA models describe a process model of the aircraft development processes and support cost analysis not only from the static perspective but also from the dynamic perspective. The content of the paper is very well illustrated through figures and tables. The fact that at the end of the conclusion the authors present a summary of their contributions also makes a good impression.

I only recommend the authors to move the text with their plans for future work in the field from Section 5 to the conclusion. 

Author Response

Thanks for your suggestion, I have put the text of the future work in the conclusion.

Reviewer 2 Report

The paper does not seem to present a new modeling technique or modeling approach, thus we can assume the new content is in the application to the real case f aircraft production, thus the title should belong to this field. Please review the proposed title.

If the paper is a case study, please underline this, change the title, and focus more on the case study, less on the behind theory if not changed considering previous works.

The paper should focus on advancements, just referring to previous studies, or at least the authors should properly underline if the paragraph is new or not in the scientific panorama: paragraph 3 does not seem to provide any new content to the previous:

  • Ding, J.; Reniers, M.; Lu, J.; Wang, G.; Feng, L.; Kiritsis, D. Integration of modeling and verification for system model based on 555 KARMA language. Proceedings of the 18th ACM SIGPLAN International Workshop on Domain-Specific Modeling, 2021, pp. 556 41–50.
  • Wang, G. Wang, J. Lu, C. Ma, Ontology supporting model-based systems engineering based on a gopprr approach, in: WorldCIST, 2019, pp. 426–436. https://doi.org/10.1007/978-3-030-16181-1_40

Please better explain the improvements and the differences with previous related works.

No validation of simulation is provided; please explain this choice.

Author Response

Thanks for your suggestion, I answer your remarks separately:

(1)I agree with your assumption, the purpose of this article is to display the capability of formal process modeling, static analysis and dynamic analysis in the specific domain through the case of aircraft development. So I change the title to "Semantic Modeling Approach Supporting Process Modeling and Analysis in aircraft development".

(2) Ding's paper you pointed out focuses on theoretical research, which proposes a dynamic analysis. This paper is combined with formal modeling, static analysis, and dynamic analysis, and applied to the development of the aircraft system in order to verify the feasibility and applicability of the method.  I add Ding's research content in the sub-sub-section "Dynamic analysis for process model" to indicate the contribution to this paper. Wang's paper is about the ontology based on the GOPPRR method. This article is about the formal modeling language developed by the GOPPRRE method, these are the essential difference. I add an explanation in the sub-section “Overview of the semantic approach”.

(3) This article only involves static analysis and dynamic analysis in the system modeling stage. After physical architectural models, the simulation stage is necessary, but this paper does not involve research and application of this stage.

Reviewer 3 Report

The paper presents an approach for using semantic modeling in the process modeling development process and supporting the static analysis of costs. The approach is illustrated with a case study, which represents the process modeling and analysis of aircraft avionics development process. 

I have the following remarks on the presented text and development:

1). Some figures precede their citation and explanation in the text, which is incorrect (these are figures 1, 2, 5). 

2). The model shown in Figure 4 uses two classes with the same name. That is a mistake.
3). Some of the figures presented are difficult to read. If possible, improve the quality of figures 1, 2, 3, 5, 7, 10.

Author Response

Thanks for your suggestion, I answer your remarks separately:
(1) I have put the figures in the right position.
(2) I have amended the mistake in Figure 4.
(3) I have improved the quality of figures 1, 2, 3, 5, 7, 10.

Round 2

Reviewer 2 Report

I think it would be better to underline the advances presented in the paper more than the revision did. 

Beside this, the paper is very interesting.

Author Response

Thanks for your suggestion, again. I add some descriptions to underline the advances of the research in the conclusion.